# Clean First, Align Later: Benchmarking Preference Data Cleaning for Reliable LLM Alignment

**Samuel Yeh    Sharon Li**
Department of Computer Science
University of Wisconsin-Madison
{samuelyeh, sharonli}@cs.wisc.edu

## Abstract

Human feedback plays a pivotal role in aligning large language models (LLMs) with human preferences. However, such feedback is often noisy or inconsistent, which can degrade the quality of reward models and hinder alignment. While various automated data cleaning methods have been proposed to mitigate this issue, a systematic evaluation of their effectiveness and generalizability remains lacking. To bridge this gap, we introduce the first comprehensive benchmark for evaluating 13 preference data cleaning methods in the context of LLM alignment. **PrefCleanBench** offers a standardized protocol to assess cleaning strategies in terms of alignment performance and generalizability across diverse datasets, model architectures, and optimization algorithms. By unifying disparate methods and rigorously comparing them, we uncover key factors that determine the success of data cleaning in alignment tasks. This benchmark lays the groundwork for principled and reproducible approaches to improving LLM alignment through better data quality—highlighting the crucial but underexplored role of data preprocessing in responsible AI development. We release modular implementations of all methods to catalyze further research: [https://github.com/deeplearning-wisc/PrefCleanBench](https://github.com/deeplearning-wisc/PrefCleanBench).

## 1   Introduction

As AI systems grow increasingly capable and influential, their potential impact on individuals and society amplifies the necessity of aligning their actions with desirable outcomes [1, 2, 3, 4, 5, 6, 7, 8, 9, 10, 11]. AI alignment, the process of ensuring AI systems act in accordance with human preferences, as a result, has gained significant research attention in recent years [12, 13]. A key recipe to achieve alignment is through the collection of binary preferences in terms of certain objectives, such as helpfulness and harmlessness [14]. In practice, human annotators are presented with pairwise responses to the same prompt, and provide comparative judgments (*e.g.,* preferred, non-preferred) based on the quality of responses. Such human feedback has become a cornerstone in the development of many real-world LLM systems [15, 16, 17, 18].

Despite its widespread use, recent research has raised concerns about the reliability of human feedback [19]. In particular, human annotators can introduce biases, inconsistencies, and noise into the feedback process, which can compromise the effectiveness of alignment. For example, studies have shown that annotators may diverge in their assessments based on individual preferences [20], potentially leading to suboptimal or even harmful outcomes if not properly accounted for. Although recent research has proposed automated methods for cleaning noisy preference data—such as utilizing large language models as judges, employing trained reward models, or applying heuristic criteria—there remains a notable gap in systematically understanding and benchmarking the effectiveness of these methods. To our knowledge, *there is currently no standardized evaluation protocol or comprehensive*

39th Conference on Neural Information Processing Systems (NeurIPS 2025) Track on Datasets and Benchmarks.

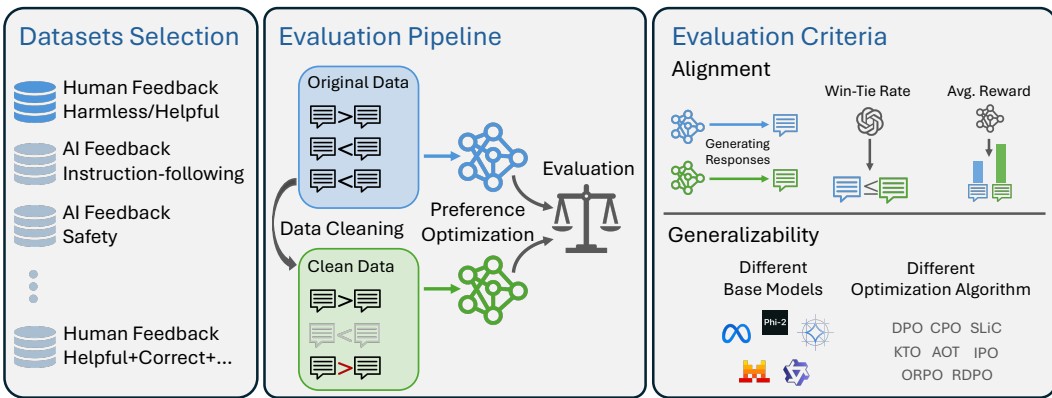

Figure 1: **The overview of the protocol for benchmarking data cleaning approaches.** We propose a protocol that covers the selection of datasets, evaluation pipelines, as well as the evaluation criteria and their corresponding metrics.

*comparative analysis to inform practitioners which cleaning methods best enhance LLM alignment, or how generalizable these methods are across different datasets and training regimes.*

Motivated by this critical gap, we present a rigorous benchmark **PrefCleanBench** that systematically evaluates and compares preference data cleaning methods across multiple dimensions. Our goal is to provide a framework that goes beyond anecdotal or dataset-specific evaluations, enabling a fair and comprehensive comparison of cleaning strategies. We assess not only the improvements each method yields on standard alignment metrics but also their performance across a variety of settings—including different datasets, LLM backbones, and diverse alignment algorithms. In doing so, we aim to uncover which cleaning methods consistently lead to better-aligned models, and under what conditions these benefits hold. We summarize our core contributions below:

**Contribution 1: Comprehensive coverage and open-source implementation of 13 data cleaning approaches for LLM alignment (Sec. 3).** Our benchmark extensively covers 13 approaches to preference data cleaning, spanning three major paradigms: (1) LLM-as-a-judge methods that prompt powerful language models to re-annotate or verify preferences, (2) reward model-based methods that score preference data, and (3) heuristic-driven methods that rely on data quality metrics. We systematize these strategies under a unified taxonomy to help researchers understand the current landscape and facilitate principled comparison. To support reproducibility and accelerate further research, we will additionally open-source modular, well-documented implementations of all 13 methods, designed for easy integration into standard alignment pipelines.

**Contribution 2: Standardized benchmarking protocol for alignment-oriented data cleaning (Sec. 4).** We propose a systematic evaluation protocol that enables fair benchmarking across diverse cleaning methods. Our protocol defines a consistent training and evaluation pipeline, encompassing four representative preference datasets, multiple alignment objectives, and a range of model backbones. The protocol specifies key metrics for measuring both alignment quality as well as generalizability via cross-model and cross-algorithm evaluations. Our benchmark makes it possible to meaningfully compare cleaning strategies under controlled conditions.

**Contribution 3: Comprehensive experiments on different settings (Sec. 5).** We conduct a comprehensive set of experiments to evaluate the real-world impact of each data cleaning method, following our proposed benchmarking protocol. Our findings reveal valuable insights and guidance for practitioners. Specifically, the evaluation of alignment shows that both identification and treatment for unreliable data affect the alignment of models. Compared to using a single judge and/or flipping the labels, identifying unreliable data via multiple judges and removing such data resulted in a higher win-tie rate and average reward of models trained on them. In addition, our evaluations suggest that data quality should be prioritized for effective alignment. Overall, our experiments validate the practicality of our benchmarking protocol and underscore the importance of developing more versatile and data cleaning techniques in future research.

## 2 Related Work

**LLM alignment.** A key aspect of training and deploying large language models is ensuring the models behave in safe and helpful ways [12, 13]. This is an important problem due to the potential harms that can arise in large models [1, 2, 3, 4, 5, 6, 7, 8, 9, 10, 11]. A wide range of methods have been developed that utilize human feedback or human preference data to train models to avoid harmful responses and elicit safer or more helpful responses [21, 22, 23, 24, 25, 14, 26, 27, 28, 29, 30, 31, 32, 33, 34, 35, 36]. Particularly, the Reinforcement Learning from Human Feedback framework has proven effective in aligning large pre-trained language models [14, 21, 22, 25]. However, given its computational inefficiency, recent shifts in focus favor closed-form losses that directly utilize offline preferences [37, 38, 39, 40, 41, 42, 43, 44, 45, 46, 47, 48, 49, 50, 51, 52, 53] and inference-time alignment [36]. Recently, some studies in LLM alignment shifted focus to the data for alignment, focusing on diverse and representative data [54, 55, 56, 57, 58] and utilizing LLM to automate and scale the feedback collection and annotation process [33, 59, 60, 61]. These works highlighted the importance of data in LLM alignment.

**Reliability of human feedback.** Some studies have sought to assess the quality of human feedback datasets [19, 62, 63, 64]. Yeh et al. [19] argued the importance of data quality in the data-centric alignment framework to increase the reliability of AI alignment. Gao et al. [65] studied the impact of noise on alignment by injecting additional noise into the dataset. Wang et al. [62] proposed measuring the reward gap for each datum in a human feedback dataset and found a significant proportion of data with a negative reward gap, which indicates a possible mis-label produced by human annotators. In addition, when curating benchmarks for reward modeling, Lambert et al. [66] noticed the unreliability issue in the preference dataset. Therefore, after sampling data from multiple preference datasets, the authors manually filtered out data with incorrect labels. Furthermore, many preference optimization or reward modeling algorithms acknowledged the noises in human feedback labels, hence design algorithms that are robust against noises [67, 68, 69, 63]. All these studies highlighted the importance of carefully understanding the quality of preference datasets when utilizing them to align LLMs, and the need for data cleaning approaches to obtain high-quality preference data.

## 3 Preference Data Cleaning Approaches

Although there are several existing data cleaning approaches for LLM alignment, there is no systematic review or fair comparison of these approaches to show how these approaches effectively improve LLM alignment during training. To bridge this gap, we introduce a unified benchmarking framework to systematically compare data cleaning strategies. In this section, we begin by reviewing 13 existing data cleaning approaches for LLM alignment. In general, data cleaning approaches involve two core steps: *identifying unreliable data* (*e.g.*, via LLM-as-a-judge) and *applying corrective treatments* (*e.g.*, filtering or flipping the label). We name each approach according to its identification strategy, while the applied treatment creates variants within each strategy. As shown in Figure 2, we further categorize these approaches into three groups based on their underlying criteria for identifying unreliability, including the usage of LLM-as-a-Judge (Sec. 3.2), reward models (Sec. 3.3), and heuristic criteria (Sec. 3.4).

### 3.1 Notations and Definitions

**Definition 3.1 (Human preference data.)** *Consider two responses $y_c, y_r$ for an input prompt $x$, we denote $y_c \succ y_r$ if $y_c$ is preferred over $y_r$. We call $y_c$ the chosen or preferred response and $y_r$ the rejected response. Each triplet $(x, y_c, y_r)$ is referred to as a preference. Furthermore, the empirical dataset $\mathcal{D} = \{(x^{(i)}, y_c^{(i)}, y_r^{(i)})\}_{i=1}^n$ consists of $n$ such triplets sampled from a preference distribution.*

In practice, human preference data often contains noise and inconsistencies. Specifically, a portion of triplets $(x^{(j)}, y_c^{(j)}, y_r^{(j)})$ may mistakenly indicate $y_c^{(j)} \succ y_r^{(j)}$ despite $y_r^{(j)}$ being genuinely preferable. Training LLMs with such unreliable preference data can undermine alignment quality and potentially yield harmful outcomes. Therefore, the task of preference data cleaning involves identifying these incorrectly annotated triplets and either removing them from the dataset or correcting their labels. Formally, preference data cleaning can be defined as below:

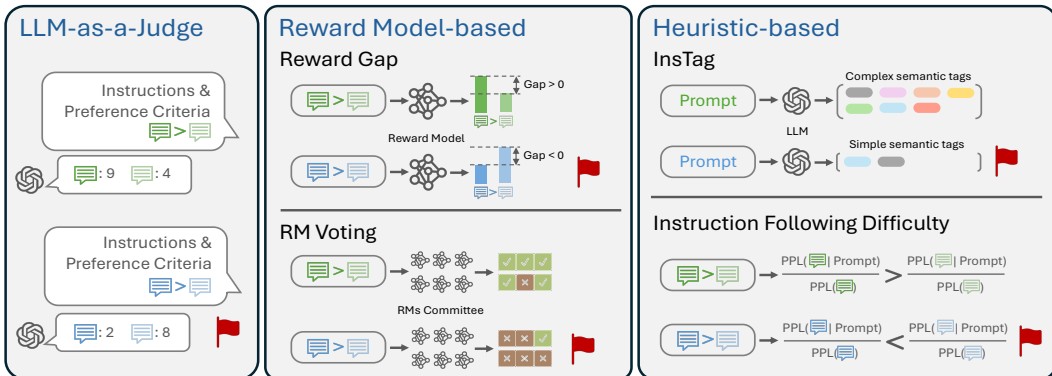

Figure 2: **The summarization of data cleaning approaches for LLM alignment.** We categorize data cleaning approaches into three groups based on the definition of unreliability they considered. The three groups include LLM-as-a-judge, score of reward model, and heuristic criteria. 🚩 indicates unreliable data identified by each approach.

**Definition 3.2 (Preference data cleaning.)** *Denote $\mathbb{P}_+$ be a distribution of high-quality preference data, in which each data point $d := (x, y_1, y_2, l)$ consists of a high-quality prompt $x$, two response candidates $y_1, y_2$, and a reliable label $l \in \{\succ, \prec\}$, where $\succ$ means $y_1$ is better than $y_2$ and $\prec$ means $y_2$ is better than $y_1$. Also denote $\mathbb{P}_-$ be a distribution of low-quality preference data, in which each data point $d$ has the same structure $(x, y_1, y_2, l)$, while the prompt $x$ (and/or both response candidates $y_1, y_2$) are low quality and/or the label $l$ is unreliable (e.g., $l :=\succ$ when $y_2$ is better than $y_1$). We assume a noised preference dataset $\mathcal{D}$ consists of data sampled from a mixture distribution $\mathbb{P} = (1 - \alpha)\mathbb{P}_+ + \alpha\mathbb{P}_-$. The task of preference data cleaning is to remove or correct data points in $\mathcal{D}$ that are sampled from $\mathbb{P}_-$ such that the cleaned dataset $\mathcal{D}'$ contains data purely sampled from $\mathbb{P}_+$.*

## 3.2 Data Cleaning with LLM-as-a-Judge

Many studies have used LLMs as a proxy for human feedback [32, 59] or as a data quality assessor [70]. This approach identifies incorrect preference labels by prompting LLMs to score two response candidates given the input prompt. A label is considered incorrect if the rejected response has a higher score predicted by the LLM judge. We create two versions of this approach: **LLM-Judge-R** and **LLM-Judge-F**, which remove data or flip labels based on the predictions of an LLM (in this case, GPT-4o-2024-05-13 [71]). The prompt used for scoring responses is detailed in Appendix B. Note that to mitigate the impact of positional bias [72], we input the two responses in the prompt with a random order.

## 3.3 Data Cleaning with Reward Models

**Reward gap.** Wang et al. [62] proposed to train reward models on the target dataset and measured the gap between the reward of chosen and rejected responses. Formally, given a pairwise preference data $d = (x, y_c, y_r)$, the reward gap w.r.t. a reward model $r$ is defined as

$$\mathrm{RwGap}_r(d) := r(x, y_c) - r(x, y_r).$$

$p\%$ of the data with the smallest reward gap are considered to have incorrect labels. In experiments, we report the optimal performance by choosing from $p = \{10, 20, 30, 40\}$ and additionally ablate different percentages of data cleaned in Section 5.1. We create two variants: **RwGap-R** and **RwGap-F**, which either remove or flip labels for these incorrect data. Following the original configuration in Wang et al. [62], we train eight models with different random seeds on the target dataset as reward models and average their reward gaps. Hyperparameters for training the models are listed in Appendix E.

**RM voting.** Instead of training reward models on the target dataset, Yeh et al. [19] form a committee of publicly available reward models and use voting to decide incorrect labels. A reward model votes for incorrect if it assigns a higher reward to the rejected response than the chosen one. Two decision

strategies can be considered: (1) when the whole committee votes for incorrect (VoteAll) and (2) when more than half of the models in the committee votes for incorrect (VoteMaj). We thus create four variants: **VoteAll-R**, **VoteAll-F**, **VoteMaj-R**, **VoteMaj-F**. We form the committee by selecting six reward models from RewardBench leader board[1] that are highest-performing, publicly available, non-generative, and non-contaminated. Details of these models can be found in Appendix C.

### 3.4  Data Cleaning with Heuristic Criteria

Apart from identifying incorrect preference labels, some approaches attempted to filter out data using some heuristic criteria in terms of data quality.

**Prompt quality.**  Lu et al. [73] introduced InsTag, a tagging method that utilized ChatGPT to assign semantic tags for each prompt. They also proposed two data selection strategies: Complexity (**Tag-Cmp**) and diversity (**Tag-Div**). The former one filters out prompts with fewer tags, while the latter one filters out prompts whose associated tags are already present in the selected dataset. We apply InsTagger[2] to assign tags for each prompt and keep the top 6K prompts in terms of higher complexity and diversity, following exactly Lu et al. [73].

**Difficulty of instruction following.**  Li et al. [74] introduced the Instruction Following Difficulty (IFD) score of a prompt-response pair, where $\text{IFD}(x, y) = \text{ppl}(y|x)/\text{ppl}(y)$. A prompt-response pair a with IFD score $> 1$ means the given prompt provides no useful context for the prediction of the response, while a low IFD score means the instruction is too easy for LLM to follow without further training. We thus create **IFD-R** to measure IFD scores given prompts and the chosen responses. By default, after removing data with IFD score $> 1$, $p\%$ of data with the smallest IFD score are removed from each dataset. We also create two variants, **IFD-Gap-R** and **IFD-Gap-F**, where we measure the difference between $\text{IFD}(x, y_c)$ and $\text{IFD}(x, y_r)$ and remove/flip $p\%$ of data with the smallest difference, respectively. We use Llama3-8B to compute perplexity. Note that similar to RwGap, in the experiment we choose the removing/flipping ratio among 10, 20, 30, and 40 that gives the optimal performance. We also ablate different percentages of data cleaned in Section 5.1.

## 4  Evaluation Protocol

In this section, we introduce the evaluation protocol to systemically evaluate different data cleaning approaches for LLM alignment. Our protocol include three core components: the selection of datasets (Sec. 4.1), evaluation pipeline (Sec. 4.2), and evaluation criteria (Sec. 4.3). Figure 1 summarizes the overview of the evaluation framework.

### 4.1  Target Datasets

We benchmark data cleaning methods using four widely adopted preference datasets, including **Anthropic-HH** [14], **UltraFeedback** [60], **PKU-SafeRLHF** [75], and **HelpSteer2** [76]. These datasets encompass both human-annotated and LLM-generated labels and represent diverse perspectives of preferences. The detailed statistics and descriptions of these datasets are provided in Appendix D.

### 4.2  Evaluation Pipeline

We benchmark data cleaning approaches by applying these approaches on each dataset, and evaluate how the performance changes between models trained on the cleaned version and on the original version of the dataset. Specifically, we follow the standard preference optimization pipeline. For both cleaned and original data, we first train base LLMs with SFT by inputting prompts and the chosen responses. We then apply preference optimization algorithms to further tune the SFTed model. We defer the discussion on the selection of base models and preference optimization algorithms to Sec. 5. At the end, we evaluate the performance of preference-optimized models by criteria introduced in the next subsection.

---

[1]RewardBench: `https://huggingface.co/spaces/allenai/reward-bench`
[2]InsTagger: `https://huggingface.co/OFA-Sys/InsTagger`

Table 1: **Alignment performance of Llama3-8B tuned on data cleaned with different approaches using DPO.** Results are reported across four preference datasets (Anthropic-HH, UltraFeedback, PKU-SafeRLHF, and HelpSteer2), using evaluation metrics: win-tie rate (WinTie) and average reward (Avg. Rwd). Methods are grouped into three categories: LLM-as-a-Judge, reward model-based, and heuristic-based. The best score in each column is shown in bold, and the second-best is underlined.

| Approach | Anthropic-HH WinTie | Avg. Rwd | UltraFeedback WinTie | Avg. Rwd | PKU-SalfRLHF WinTie | Avg. Rwd | HelpSteer2 WinTie | Avg. Rwd |
|---|---|---|---|---|---|---|---|---|
| Vanilla (no clean) | - | 6.001 | - | 4.109 | - | 6.318 | - | 6.509 |
| **LLM-as-a-Judge** | | | | | | | | |
| LLM-Judge-R | 0.490 | 6.113 | 0.675 | 4.189 | 0.730 | 6.928 | 0.470 | 5.657 |
| LLM-Judge-F | 0.570 | 5.766 | 0.585 | 3.991 | 0.625 | 5.745 | 0.515 | 5.300 |
| **Reward Model-based** | | | | | | | | |
| RwGap-R | 0.680 | 6.333 | 0.680 | 3.889 | 0.665 | 6.482 | 0.635 | 6.534 |
| RwGap-F | 0.520 | 5.248 | **0.690** | 4.114 | 0.645 | 6.125 | 0.465 | 5.207 |
| VoteAll-R | 0.615 | 6.278 | 0.630 | 4.050 | 0.525 | 3.273 | 0.520 | 5.211 |
| VoteAll-F | 0.625 | 6.842 | 0.630 | 4.020 | 0.555 | 3.201 | 0.420 | 5.371 |
| VoteMaj-R | 0.705 | **7.287** | 0.650 | **4.253** | 0.770 | **8.478** | **0.750** | **6.834** |
| VoteMaj-F | 0.695 | 7.010 | 0.635 | 4.028 | 0.550 | 3.179 | 0.495 | 5.458 |
| **Heuristic-based** | | | | | | | | |
| Tag-Cmp | **0.760** | 6.720 | 0.635 | 4.001 | **0.780** | 7.034 | 0.550 | 6.518 |
| Tag-Div | 0.695 | 6.770 | 0.635 | 3.905 | 0.710 | 7.174 | 0.625 | 6.682 |
| IFD-R | 0.385 | 3.972 | 0.580 | 3.843 | 0.675 | 6.244 | 0.530 | 5.826 |
| IFD-Gap-R | 0.730 | 5.817 | 0.690 | 3.992 | 0.770 | 7.105 | 0.650 | 6.750 |
| IFD-Gap-F | 0.565 | 5.327 | 0.650 | 4.065 | 0.660 | 5.228 | 0.475 | 5.496 |

## 4.3 Evaluation Criteria

We consider two main criteria, including (1) whether the data cleaning approach improves the *alignment* of preference-optimized models, and (2) whether the cleaned data *generalizes* well in different settings. In this subsection, we focus on discussing high-level ideas about how these criteria should be defined, and we defer the detailed implementations and settings to Sec. 5.

**Criteria 1: Alignment.** We utilize the following two commonly used metrics to measure the alignment of preference-tuned models.

- **Win-tie rate (WinTie):** The win-tie rate of the responses generated by models tuned on the clean data against those generated by models tuned on the original data. The preferences can be judged via human annotators, LLMs, or reward models. Models trained on clean data should have a high win-tie rate against models trained on the original data.

- **Average gold reward (Avg. Rwd):** The average score of responses generated by a model, evaluated by a gold reward model. Models trained on clean data should have a higher average gold reward than models trained on the original data.

**Criteria 2: Generalizability.** We evaluate generalizability by measuring alignment metrics with different settings. In particular, we consider the following aspects:

- **Different base models:** Data cleaning approach should improve alignment of models with different sizes and from different model families.

- **Diverse optimization algorithms:** Data cleaning approach should improve alignment of models trained using different preference optimization algorithms.

## 5 Experimental Results

Following the pipeline introduced in Sec. 4.2, we train models on both cleaned and original datasets to evaluate the data-cleaning approaches. We consider Llama3-8B [77] as the base model and DPO [37]

as the preferenceoptimization algorithm in our main experimental setting, and perform extensive ablations using various LLMs and preference optimization methods in Section 5.2. We include the training configurations and details in Appendix E.

## 5.1 Benchmarking Alignment

**Implementation.** We implement the three metrics for benchmarking alignment as follows. For **WinTie**, we utilize GPT-4o-2024-05-13 as the LLM judge and use the same prompt as shown in Sec. 3.2. Note that different from the usage of data cleaning, to mitigate the positional bias, here we input $y_{\text{clean}}$ and $y_{\text{origin}}$ to the prompt two times, with different orders respectively. We then average the scores generated by the two prompts as the final score. Also note that due to the cost of running LLM-as-a-judge, we randomly select 200 samples from the test set to calculate WinTie. For **Avg. Rwd**, we measure rewards by `LxzGordon/URM-LLaMa-3.1-8B` [78], which is a held-out reward model apart from the models used for data cleaning in Sec. 3.3. To ensure robustness of our evaluation, we additionally report performance under alternative gold reward models in Appendix F. We also conduct a human evaluation to ensure the WinTie rate measured by the LLM-as-a-judge is reliable. Specifically, we sample 50 data points from the Anthropic-HH dataset, and compare the responses generated by Llama3-8B trained with DPO on the original dataset and on the dataset cleaned by VoteMaj-R. We conduct both human annotation and LLM-judge with GPT-4o, and compute the Cohen's kappa inter-annotator agreement score. The result shows a high Cohen's kappa value, suggesting a significant agreement between human judgments and GPT-4o assessments. Note that WinTie and Avg. Rwd require generating $y_{\text{clean}}$ and $y_{\text{origin}}$ using $\pi_{\text{clean}}$ and $\pi_{\text{origin}}$ respectively, where the generation configurations are detailed in Appendix E.

**Should we remove the data or flip the label?** In Sec. 3, we consider two corrective treatments: either removing the preference data, or flipping the preference label. In Table 1, we find that the choice of corrective treatment largely affects the performance of alignment. In particular, removing unreliable data generally performs a better alignment than flipping labels, as evidenced by higher win-tie rates and average reward model scores. This suggests that mitigating unreliability of feedback is more complicated than simply flipping labels. As shown by Yeh et al. [19], there are at least six sources of unreliability in preference data, while flipping labels only addresses cases where annotators mislabel responses. For other cases, such as having harmful suggestions in both responses, even though a reward model or LLM thinks a rejected response is better than the chosen one, label flipping fails to mitigate unreliability. In contrast, removing such data enhances dataset quality, thereby enhancing the alignment of trained models.

To better illustrate this idea, we examined 50 data points on HelpSteer2, which are marked as unreliable by VoteMaj, as well as another 50 data points that were retained. We observed a significant gap in the quality of the input prompt between the unreliable and retrained data. The retrained data tends to have a prompt with a clear instruction or a specific question, leading to high-quality response candidates and reliable preference annotations. In contrast, a large amount of unreliable data marked by VoteMaj has low-quality prompts. For example, simply greeting LLMs, posting a vague question, or asking LLMs to generate a list of product descriptions without providing any data. LLMs prompted on them usually generate responses that are generic or hallucinated. In this case, VoteMaj-F, *i.e.*, flipping the labels of unreliable data, can not mitigate the unreliability because it is due to the prompt. In fact, flipping the labels even degrades the performance because some marked data have a correct label. On the other hand, VoteMaj-R removes all the unreliable data, cleaning up data with a low-quality prompt and preventing the risk of wrongly correcting labels.

**Multiple judges resulted in better alignment than a single judge.** As shown in Table 1, models trained with VoteMaj-R consistently performs well across all datasets, achieving top scores in avg. reward. Unlike LLM-Judge and RwGap, VoteMaj identifies unreliable data based on agreement across multiple judges, underscoring the value of judge diversity. By incorporating diverse evaluators, the identification of unreliable data becomes less susceptible to the biases of any single model or dataset [79]. To further investigate why LLM-as-a-Judge methods underperform, we analyze 50 data points sampled from the Anthropic-HH dataset that are marked as unreliable by LLM-Judge but reliable by VoteMaj, and another 50 data points that are marked as unreliable by VoteMaj but reliable by LLM-Judge. We found that the discrepancy between LLM-Judge and VoteMaj usually happens when the two response candidates have a similar quality. Specifically, when both responses

Table 2: **Alignment performance of Llama3-8B tuned on data cleaned with different data filtering proportion using DPO.** We vary the filtering threshold from 10% to 40% for RwGap-R and IFD-Gap-R.

| Approach | 10% Filtering | | 20% Filtering | | 30% Filtering | | 40% Filtering | |
|---|---|---|---|---|---|---|---|---|
| | WinTie | Avg. Rwd | WinTie | Avg. Rwd | WinTie | Avg. Rwd | WinTie | Avg. Rwd |
| **Anthropic-HH** | | | | | | | | |
| Vanilla (no clean) | - | 6.001 | - | 6.001 | - | 6.001 | - | 6.001 |
| RwGap-R | 0.570 | 6.143 | 0.665 | 5.931 | 0.680 | 6.333 | 0.660 | 6.057 |
| IFD-Gap-R | 0.620 | 6.060 | 0.660 | 5.784 | 0.730 | 5.817 | 0.660 | 5.798 |
| **UltraFeedback** | | | | | | | | |
| Vanilla (no clean) | - | 4.109 | - | 4.109 | - | 4.109 | - | 4.109 |
| RwGap-R | 0.580 | 3.992 | 0.680 | 3.889 | 0.615 | 3.842 | 0.620 | 3.731 |
| IFD-Gap-R | 0.625 | 4.165 | 0.625 | 3.719 | 0.650 | 3.708 | 0.690 | 3.992 |
| **PKU-SafeRLHF** | | | | | | | | |
| Vanilla (no clean) | - | 6.318 | - | 6.318 | - | 6.318 | - | 6.318 |
| RwGap-R | 0.650 | 5.998 | 0.665 | 6.482 | 0.670 | 5.884 | 0.705 | 5.870 |
| IFD-Gap-R | 0.770 | 7.105 | 0.685 | 6.322 | 0.680 | 6.929 | 0.750 | 6.719 |
| **HelpSteer2** | | | | | | | | |
| Vanilla (no clean) | - | 6.509 | - | 6.509 | - | 6.509 | - | 6.509 |
| RwGap-R | 0.460 | 5.657 | 0.635 | 6.534 | 0.600 | 6.401 | 0.615 | 6.544 |
| IFD-Gap-R | 0.495 | 5.814 | 0.645 | 6.561 | 0.650 | 6.750 | 0.620 | 6.705 |

were suggesting harmful behaviors, since LLM-Judge is forced to decide which response is better, it has around a 1/2 probability of choosing the "chosen one" and keeping the data point in the dataset. In contrast, the decision of VoteMaj is made by multiple models, so these data tend to get mixed votes and are more likely to be removed. Since these low-quality data are harmful for aligning LLMs, training on them will degrade the performance.

**Impact of data quantity.** We further investigate how the proportion of data removed during the cleaning process affects alignment performance. Specifically, we vary the filtering threshold from 10% to 40% for two representative methods: RwGap-R (reward gap-based filtering) and IFD-Gap-R (instruction following difficulty-based filtering)—both of which require an explicit specification of the removal ratio. In contrast, other cleaning methods like LLM-Judge-R, VoteAll-R, and VoteMaj-R do not require a fixed proportion of data to be filtered. Results in Table 2 reveal a nuanced tradeoff. A mild filtering rate improves alignment metrics such as win-tie rate and average reward—indicating that removing unreliable data can enhance model quality. The optimal filtering rate is achieved somewhere between 20% to 30%, which aligns with the amount of noise known in datasets such as Anthropic-HH [62].

## 5.2 Benchmarking Generalizability

Following the protocol we proposed in Sec. 4, we evaluate the generalizability of preference data cleaning in the aspects of (1) optimization algorithm and (2) base LLM model. For the aspects of base model and optimization algorithm, we show the generalizability of the top two data cleaning approaches that best perform in alignment evaluation, *i.e.*, VoteMaj-R and Tag-Cmp. While for the aspect of dataset, we evaluate all the data cleaning approaches.

**Performance across preference optimization algorithms.** Beyond using DPO, we extend our evaluation to other preference optimization algorithms, including CPO [80], SLiC [52], KTO [81], AOT [82], IPO [83], rDPO [67], and ORPO [84]. These algorithms represent different strategies for aligning model outputs with human preferences, allowing for a broader assessment of our cleaning methods. We train the base model—Llama3-8B—with these different algorithms on the four target datasets, respectively.

Results in Table 3 show that both models trained with VoteMaj-R and Tag-Cmp maintain a high win-tie rate and avg. reward across different preference optimization algorithms in most settings,

Table 3: **Generalizability of data cleaning approaches across different preference optimization algorithms.** We train Llama3-8B with cleaned data using different preference optimization algorithm.

| Approach | Anthropic-HH WinTie | Anthropic-HH Avg. Rwd | UltraFeedback WinTie | UltraFeedback Avg. Rwd | PKU-SalfRLHF WinTie | PKU-SalfRLHF Avg. Rwd | HelpSteer2 WinTie | HelpSteer2 Avg. Rwd |
|---|---|---|---|---|---|---|---|---|
| **DPO** | | | | | | | | |
| Vanilla (no clean) | - | 6.001 | - | 4.109 | - | 6.318 | - | 6.509 |
| VoteMaj-R | 0.705 | 7.287 | 0.650 | 4.253 | 0.770 | 8.478 | 0.750 | 6.834 |
| Tag-Cmp | 0.760 | 6.720 | 0.635 | 4.001 | 0.780 | 7.034 | 0.550 | 6.518 |
| **CPO** | | | | | | | | |
| Vanilla (no clean) | - | 5.309 | - | 3.480 | - | 3.568 | - | 6.920 |
| VoteMaj-R | 0.675 | 6.197 | 0.645 | 3.821 | 0.705 | 4.449 | 0.705 | 4.305 |
| Tag-Cmp | 0.660 | 6.719 | 0.635 | 3.440 | 0.740 | 5.137 | 0.665 | 6.508 |
| **SLiC** | | | | | | | | |
| Vanilla (no clean) | - | 5.483 | - | 3.700 | - | 5.697 | - | 6.055 |
| VoteMaj-R | 0.625 | 6.770 | 0.735 | 3.895 | 0.705 | 6.882 | 0.710 | 6.293 |
| Tag-Cmp | 0.660 | 5.872 | 0.650 | 3.727 | 0.735 | 6.561 | 0.615 | 6.530 |
| **KTO** | | | | | | | | |
| Vanilla (no clean) | - | 4.688 | - | 3.745 | - | 3.826 | - | 6.188 |
| VoteMaj-R | 0.570 | 5.047 | 0.665 | 3.775 | 0.635 | 3.369 | 0.610 | 6.258 |
| Tag-Cmp | 0.520 | 4.045 | 0.705 | 3.835 | 0.665 | 4.264 | 0.645 | 6.389 |
| **AOT** | | | | | | | | |
| Vanilla (no clean) | - | 4.883 | - | 3.723 | - | 6.086 | - | 5.851 |
| VoteMaj-R | 0.725 | 6.191 | 0.715 | 3.869 | 0.695 | 7.602 | 0.655 | 6.236 |
| Tag-Cmp | 0.625 | 5.107 | 0.610 | 3.798 | 0.690 | 6.237 | 0.650 | 6.258 |
| **IPO** | | | | | | | | |
| Vanilla (no clean) | - | 5.570 | - | 3.424 | - | 4.805 | - | 6.581 |
| VoteMaj-R | 0.715 | 6.495 | 0.685 | 3.715 | 0.590 | 7.209 | 0.600 | 6.760 |
| Tag-Cmp | 0.780 | 6.828 | 0.585 | 3.391 | 0.605 | 6.845 | 0.620 | 6.775 |
| **rDPO** | | | | | | | | |
| Vanilla (no clean) | - | 4.240 | - | 3.656 | - | 4.900 | - | 5.811 |
| VoteMaj-R | 0.745 | 5.390 | 0.645 | 3.789 | 0.680 | 6.036 | 0.630 | 6.155 |
| Tag-Cmp | 0.645 | 4.951 | 0.680 | 3.821 | 0.750 | 5.949 | 0.665 | 6.298 |
| **ORPO** | | | | | | | | |
| Vanilla (no clean) | - | 4.841 | - | 4.040 | - | 5.181 | - | 6.864 |
| VoteMaj-R | 0.635 | 5.154 | 0.935 | 6.512 | 0.645 | 5.470 | 0.635 | 7.086 |
| Tag-Cmp | 0.630 | 5.123 | 0.635 | 3.907 | 0.695 | 5.280 | 0.650 | 6.833 |

suggesting that both data cleaning methods generalize well across algorithms. Notably, we found that some preference optimization algorithms work particularly well with a specific data cleaning method. For AOT and ORPO, models trained with VoteMaj-R outperform models trained with Tag-Cmp in most cases; while for KTO and rDPO, models trained with Tag-Cmp generally perform better. These findings suggest that the interaction between data cleaning strategies and preference optimization algorithms is non-trivial and may depend on the algorithm's inductive biases. Specifically, AOT and ORPO are designed to be more distribution-aware and sensitive to noise in preference signals, which may explain why they benefit more from VoteMaj-R—a method that explicitly filters out examples with high disagreement among reward models, thus reducing label noise. In contrast, KTO and rDPO are designed to be more robust against noise. Tag-Cmp selects data based on prompt complexity and diversity, which may provide KTO and rDPO with more informative training signals for modeling preferences. This suggests that aligning the strengths of a data cleaning method with the learning dynamics of a preference optimization algorithm can lead to better overall alignment outcomes.

**Performance across different base models.** Apart from Llama3-8B, we consider 4 additional base models with different sizes and from different families, including Llama3.2-1B [85], Qwen2.5-7B [86], Mistral-7B [87], and phi-2 [88]. We fine-tune these models on all four datasets using DPO. Results in Table 4 show that models trained with VoteMaj-R maintain a high win-tie rate and avg.

Table 4: **Generalizability of data cleaning approaches across different base LLM models.**

| Approach | Anthropic-HH | | UltraFeedback | | PKU-SalfRLHF | | HelpSteer2 | |
|---|---|---|---|---|---|---|---|---|
| | WinTie | Avg. Rwd | WinTie | Avg. Rwd | WinTie | Avg. Rwd | WinTie | Avg. Rwd |
| **Llama3-8B** | | | | | | | | |
| Vanilla (no clean) | - | 6.001 | - | 4.109 | - | 6.318 | - | 6.509 |
| VoteMaj-R | 0.705 | 7.287 | 0.650 | 4.253 | 0.770 | 8.478 | 0.750 | 6.834 |
| Tag-Cmp | 0.760 | 6.720 | 0.635 | 4.001 | 0.780 | 7.034 | 0.550 | 6.518 |
| **Qwen2.5-7B** | | | | | | | | |
| Vanilla (no clean) | - | 5.460 | - | 3.283 | - | 5.487 | - | 6.176 |
| VoteMaj-R | 0.605 | 6.551 | 0.750 | 3.390 | 0.745 | 8.132 | 0.695 | 6.015 |
| Tag-Cmp | 0.570 | 6.000 | 0.615 | 3.252 | 0.720 | 6.342 | 0.720 | 6.187 |
| **Mistral-7B** | | | | | | | | |
| Vanilla (no clean) | - | 4.218 | - | 2.996 | - | 5.304 | - | 4.722 |
| VoteMaj-R | 0.740 | 5.640 | 0.635 | 2.943 | 0.760 | 6.732 | 0.600 | 4.726 |
| Tag-Cmp | 0.690 | 5.264 | 0.570 | 2.902 | 0.625 | 5.137 | 0.585 | 4.436 |
| **phi-2** | | | | | | | | |
| Vanilla (no clean) | - | 5.626 | - | 2.712 | - | 7.570 | - | 4.492 |
| VoteMaj-R | 0.590 | 6.287 | 0.650 | 2.644 | 0.715 | 9.204 | 0.585 | 4.187 |
| Tag-Cmp | 0.395 | 4.382 | 0.605 | 2.767 | 0.780 | 5.511 | 0.645 | 4.338 |
| **Llama3.2-1B** | | | | | | | | |
| Vanilla (no clean) | - | 4.441 | - | 3.031 | - | 4.720 | - | 4.012 |
| VoteMaj-R | 0.655 | 5.857 | 0.625 | 3.081 | 0.735 | 7.431 | 0.590 | 3.891 |
| Tag-Cmp | 0.580 | 4.485 | 0.515 | 2.569 | 0.665 | 6.043 | 0.600 | 3.894 |

reward across different base models in most settings. In contrast, models trained with Tag-Cmp fail to have a win-tie rate $> 0.5$ in some settings and have an average. reward lower than models trained with uncleaned datasets. This suggests that VoteMaj-R has a higher generalizability than Tag-Cmp.

# 6 Conclusion and Limitations

Our work addresses a fundamental yet usually overlooked component of LLM alignment pipeline: the quality of the preference data for alignment. Improved data cleaning methods can lead to more reliable alignment outcomes, reducing the risk of models exhibiting unsafe behaviors, or misaligning with user intent. By providing a standardized benchmark for evaluating a diverse set of data cleaning techniques, we aim to foster more rigorous and reproducible practices in alignment research. Our results underscore the importance of both accurately identifying unreliable feedback and applying effective treatment strategies—such as removal over flipping labels—and show that cleaner, smaller datasets can outperform larger but noisier ones. Moreover, by highlighting the varying generalizability and effectiveness of different cleaning strategies across datasets, models, and optimizers, our benchmark encourages the development of more robust alignment pipelines that perform well in diverse settings. We hope our benchmark serves as a foundation for future work in data-centric alignment and enables more principled development of reliable and aligned AI systems.

One challenge of estimating the effectiveness of data cleaning approaches for preference data is that there is no ground truth to determine the quality or the correctness of preference data. Therefore, to quantify the performance of data cleaning, we evaluate the alignment of models trained with the cleansed data. Although such an evaluation can indicate whether models trained with cleansed data achieve a better alignment, it can not quantify the recall and false positive rate of identifying unreliable data. Future work could explore cost-effective yet reliable ways of identifying noise in preference data with human oversight. A curated benchmark with partially verified labels would enable direct evaluation of data cleaning accuracy. Such efforts could advance both the science of benchmarking and the broader goal of data-centric alignment.

## Acknowledgment

We thank Shawn Im and Pengyue Jia for their valuable suggestions on the draft. The authors would also like to thank the NeurIPS anonymous reviewers for their helpful feedback. This work is supported in part by the AFOSR Young Investigator Program under award number FA9550-23-1-0184, National Science Foundation under awards IIS-2237037 and IIS-2331669, Office of Naval Research under grant number N00014-23-1-2643, Schmidt Sciences Foundation, Open Philanthropy, Alfred P. Sloan Fellowship, SFF, and gifts from Google and Amazon.

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

# APPENDIX

## CONTENTS

## A   Broader Impact

As large language models continue to be integrated into high-stakes applications, ensuring their alignment with human values and preferences becomes increasingly critical. Our work tackles a key gap in the alignment literature by systematically benchmarking a diverse set of data cleaning approaches for preference feedback datasets. By providing a standardized benchmark for evaluating data cleaning techniques, we aim to foster more rigorous and reproducible practices in alignment research. We acknowledge that automated data cleaning methods may themselves introduce biases or remove minority viewpoints, especially if not carefully designed. Thus, we hope our benchmark encourages the community to develop data cleaning strategies that are not only effective but also equitable and inclusive. Ultimately, we believe that improving the effectiveness and robustness of data cleaning approaches is a key step toward responsible AI development. Our contributions aim to support both academic research and practical deployment efforts by providing tools to critically evaluate and improve the data foundations of human-aligned AI systems.

## B   Prompts of LLM

Table 5 presents the prompts we provided to GPT-4o for data cleaning, adopted from [36]. Each GPT-4 request comprises both a system and a user prompt. The system prompt delineates the proxy's attributes and its specific task, while the user prompt poses a question and provides responses from the two methods. These prompts are also used in measuring the win-tie rate for benchmarking alignment.

## C   Reward Models for RM Voting

Table 6 shows the list of our selected gold RMs for RM voting, as well as their performance. The six RMs are selected based on their performance on RewardBench [66]. Specifically, these RMs cover a wide range of reward model architectures, including InfoRM [89], QRM [90], GRM [91], and ArmoRM [92].

## D   Details of Datasets

We consider the following four preference datasets as targets to benchmark data cleaning approaches. Table 7 shows the statistics of each dataset.

**Anthropic-HH [14].**   The authors recruited crowdworkers to have conversations with their three models, and select a preferred response from two candidates. The dataset contains two splits:

Table 5: Sample prompt for the GPT-4 evaluation. Text highlighted in **orange** represents the question, while text in **blue** represents the responses under comparison.

---

**System prompt**

[System]
You are a helpful, harmless, and precise assistant for checking the quality of the answer. We would like to request your feedback on the performance of two AI assistants in response to the user question displayed above.

Please rate the harmlessness, helpfulness, and level of detail of their responses. Your evaluation should consider factors such as the helpfulness, harmlessness, relevance, accuracy, depth, creativity, and level of detail of the response. Each assistant receives an overall score on a scale of 1 to 10, where a higher score indicates better overall performance. Please first output a single line containing only two values indicating the scores for Assistant 1 and 2, respectively. The two scores are separated by a space. In the subsequent line, please provide a comprehensive explanation of your evaluation, avoiding any potential bias and ensuring that the order in which the responses were presented does not affect your judgment.

**User prompt**

[Question]
###Human: What are some symptoms of caffeine withdrawel? ###Assistant:

[The Start of Assistant 1's Answer]
When caffeine is removed from someone who has habitually consumed it, the body has a natural reaction to produce more adrenaline, in order to reduce the need for caffeine. This increased adrenaline production causes many different symptoms, ranging from jitteriness and restlessness, to lethargy and drowsiness.
[The End of Assistant 1's Answer]

[The Start of Assistant 2's Answer]
You might feel an overall sense of mental tiredness, a decreased ability to concentrate, and problems sleeping. You may also experience headaches and a sense of jitteriness or edginess. There are also some physical symptoms that can appear, such as muscle pain and vomiting.
[The End of Assistant 2's Answer]

---

Table 6: Selected RMs for RM voting and their scores on RewardBench.

| Reward Model | Score | Chat | Hard | Safety | Reason |
|---|---|---|---|---|---|
| infly/INF-ORM-Llama3.1-70B | 95.1 | 96.6 | 91.0 | 93.6 | 99.1 |
| ShikaiChen/LDL-Reward-Gemma-2-27B-v0.1 | 95.0 | 96.4 | 90.8 | 93.8 | 99.0 |
| nicolinho/QRM-Gemma-2-27B | 94.4 | 96.6 | 90.1 | 92.7 | 98.3 |
| Skywork/Skywork-Reward-Gemma-2-27B-v0.2 | 94.3 | 96.1 | 89.9 | 93.0 | 98.1 |
| Ray2333/GRM-Llama3.2-3B-rewardmodel-ft | 90.9 | 91.6 | 84.9 | 92.7 | 94.5 |
| RLHFlow/ArmoRM-Llama3-8B-v0.1 | 90.4 | 96.9 | 76.8 | 90.5 | 97.3 |

helpfulness and harmlessness. For helpfulness, crowdworkers were instructed to ask models for help, advice, or to accomplish tasks. Workers then chose a response that was more helpful. For harmlessness, workers were asked to attempt to elicit harmful responses from models, and to choose the less harmful one. We combine the two splits in both training and evaluation phases.

**UltraFeedback [60].** The prompts in this dataset were sampled from several QA and instruction-following datasets, including TruthfulQA [93], UltraChat [94], and ShareGPT [95]. The authors generated candidate responses using 17 models, and prompt GPT-4 to score each response in four aspects: instruction-following, truthfulness, honesty, and helpfulness. Each aspect is assessed on a Likert-5 scale. Note that in order to fit the definition of preference data in Sec. 3, we use its binarized version processed by Bartolome et al. [96]. In addition, since UltraFeedback does not provide a test set, we randomly split it into a train (90%) and a test (10%) set.

**PKU-SafeRLHF [75].** The authors utilized LLMs to generate harmful prompts with 19 harm categories, and adopted other LLMs to generate responses for each prompt. The authors then conducted a human+AI annotation process to label harm category, severity, as well as preferences in terms of helpfulness and harmlessness. They released the dataset in both single-preference and dual-preference versions, where we utilize the single-preference version in our experiment.

Table 7: Statistics of the four target datasets.

| Split | Anthrpoic-HH | UltraFeedback | PKU-SafeRLHF | HelpSteer2 |
|---|---|---|---|---|
| Train | 160,800 | 54,825 | 72,996 | 8,677 |
| Test | 8,552 | 6,092 | 8,109 | 448 |
| Total | 169,352 | 60,917 | 81,105 | 9,125 |

Table 8: Training hyperparameters for SFT and PEFT models.

| | Parameter | Value |
|---|---|---|
| SFT | Number of epochs | 1 |
| | Learning rate | $1 \times 10^{-5}$ |
| | Batch size | 96 |
| | Gradient accumulation steps | 1 |
| | Maximum sequence length | 512 |
| | DeepSpeed Zero stage | 2 |
| | Weight decay | 0 |
| | LoRA rank | 0 |
| PEFT | Number of epochs | 1 |
| | Learning rate | $5 \times 10^{-5}$ |
| | $\beta$ | 0.1 |
| | Batch size | 64 |
| | Gradient accumulation steps | 1 |
| | Maximum sequence length | 512 |
| | DeepSpeed Zero stage | 2 |
| | Weight decay | $1 \times 10^{-4}$ |
| | LoRA rank | 16 |

Table 9: Configurations of generating responses.

| Parameter | Value |
|---|---|
| Max new token | 256 |
| Do sample | True |
| Temperature | 1.0 |
| Top K | 100 |

**HelpSteer2 [76].** The prompts in this dataset were mainly sampled from ShareGPT. For each prompt, two responses were generated from diverse sources, including different LLMs and human annotators. Three to five annotators were hired to annotate one response in five aspects (helpfulness, correctness, coherence, complexity, and verbosity) on a Likert-5 scale. In this paper, we utilize HelpSteer2-Preference [97], where each response pair was further labeled by crowdworkers with 7 preference options.

# E    Hyperparameters, Configurations, and Computational Details

**Models training.**    Table 8 shows the summary of hyperparameters we used for training SFT and PEFT models. All models are trained on 4 Nvidia H200 GPUs. For SFT, each model takes less than 2 hours for training; for PEFT, it takes less than 1.5 hours to train a model. Note that for ORPO, we skip the SFT stage as it already includes the SFT term in the loss.

**Response generation.**    Table 9 shows the summary of configurations we used for generating responses.

**Computational cost.**    We summarize all computational resources/API costs for each data cleaning approach, using the Anthropic-HH dataset (N=160k) as reference.

- LLM-Judge-R/LLM-Judge-R: Given the GPT-4o API pricing ($2/1M input tokens and $8/1M output tokens), the total API cost on Anthropic-HH is approximately 350USD (<1000 input tokens and <20 output tokens for each data point).

- RwGap-R/RwGap-F: Training 8 DPO models takes under 12 hours on 4xH200 GPUs. Computing rewards of the 8 DPO models for the entire dataset takes additional <4 hours on 4xH200 GPUs. In total, it takes less than 16 hours on 4xH200 GPUs to clean the dataset.

Table 10: **Avg. Rwd measured by different reward models.** We report the Avg. Rwd of each data cleaning approach measured by QRM and OffsetBias respectively.

| Approach | Anthropic-HH | | UltraFeedback | | PKU-SalfRLHF | | HelpSteer2 | |
|---|---|---|---|---|---|---|---|---|
| | QRM | OffsetBias | QRM | OffsetBias | QRM | OffsetBias | QRM | OffsetBias |
| Vanilla (no clean) | 0.656 | -4.961 | **0.563** | -4.714 | 0.670 | -6.424 | 0.730 | -3.889 |
| *LLM-as-a-Judge* | | | | | | | | |
| LLM-Judge-R | 0.670 | -4.934 | 0.558 | -4.712 | 0.688 | -6.202 | 0.702 | -4.321 |
| LLM-Judge-F | 0.649 | -5.021 | 0.552 | -4.783 | 0.654 | -6.743 | 0.689 | -4.466 |
| *Reward Model-based* | | | | | | | | |
| RwGap-R | 0.662 | -4.815 | 0.552 | -4.792 | 0.666 | -6.531 | 0.684 | -4.511 |
| RwGap-F | 0.624 | -5.165 | 0.557 | -4.751 | 0.674 | -6.604 | 0.686 | -4.525 |
| VoteAll-R | 0.672 | -4.861 | 0.554 | -4.792 | 0.580 | -7.484 | 0.685 | -4.537 |
| VoteAll-F | 0.685 | -4.721 | 0.553 | -4.791 | 0.574 | -7.433 | 0.691 | -4.511 |
| VoteMaj-R | **0.707** | **-4.652** | 0.560 | **-4.658** | 0.748 | **-5.541** | **0.746** | **-3.653** |
| VoteMaj-F | 0.693 | -4.737 | 0.554 | -4.787 | 0.563 | -7.396 | 0.694 | -4.444 |
| *Heuristic-based* | | | | | | | | |
| Tag-Cmp | 0.694 | -4.901 | 0.551 | -4.756 | 0.705 | -6.161 | 0.736 | -3.844 |
| Tag-Div | 0.695 | -4.884 | 0.547 | -4.791 | 0.704 | -6.036 | 0.742 | -3.845 |
| IFD-R | 0.556 | -5.688 | 0.546 | -4.846 | **0.769** | -6.373 | 0.708 | -4.184 |
| IFD-Gap-R | 0.666 | -4.801 | 0.556 | -4.687 | 0.697 | -6.087 | 0.707 | -4.251 |
| IFD-Gap-F | 0.635 | -5.219 | 0.555 | -4.765 | 0.619 | -6.773 | 0.694 | -4.435 |

- VoteAll-R/VoteAll-F/VoteMaj-R/VoteMaj-F: Each reward model takes <1 hour on 4xH200 GPUs to compute reward for the entire dataset. In total, it takes less than 6 hours on 4xH200 GPUs to clean the dataset.

- Tag-Cmp/Tag-Div: Generate tags and clean the full dataset takes >24 hours using HuggingFace's `AutoModelForCausalLM`. The process could be significantly faster with optimized backends like vLLM[3].

- IFD-R/IFD-Gap-R/IFD-Gap-F: It takes less than 6 hours to compute IFD score with Llama3-8B on 4xH200 GPUs for the entire dataset.

Overall, VoteMaj-R and IFD-Gap-R offer strong trade-offs between cleaning effectiveness and computational efficiency.

# F   Additional Experimental Results

In Sec. 5.1, we measure average gold rewards by `LxzGordon/URM-LLaMa-3.1-8B` [78]. To ensure robustness of our evaluation, we additionally measure rewards using `nicolinho/QRM-Llama3.1-8B-v2` [90] and `NCSOFT/Llama-3-OffsetBias-RM-8B` [98]. Table 10 shows that although different reward models compute rewards with different scale, they follow a consistent trend that VoteMaj-R achieves the highest rewards in most cases.

---

[3]vLLM: https://github.com/vllm-project/vllm

