# OpenReview forum: "Clean First, Align Later: Benchmarking Preference Data Cleaning for Reliable LLM Alignment"
_NeurIPS.cc/2025/Datasets_and_Benchmarks_Track — NeurIPS 2025 Datasets and Benchmarks Track poster_

### Official Review · Reviewer_1XUo · 2025-06-01

**Rating:** 4
**Confidence:** 2

**Summary:**

PrefCleanBench addresses the issue of noisy human feedback hindering LLM alignment evaluating 13 different preference data cleaning methods. It provides a standardized protocol, open-source implementations, and assesses methods across diverse datasets, models, and optimization algorithms. The research uncovers key factors for successful data cleaning, highlighting that removing unreliable data (identified by multiple judges) generally outperforms label flipping, and underscores the critical role of data preprocessing in responsible AI development.

**Dataset Code Accessibility:**

Yes

**Ethical Considerations:**

No, there are no or only very minor ethics concerns

**Final Justification:**

4: Borderline accept

**Limitations Weaknesses:**

1. The paper should discuss the varying computational resources required by different methods and the crucial trade-off between effectiveness and cost for practical deployment.
2. Using GPT-4o as a gold standard may introduce new biases. Human evaluation is needed to validate the reliability of the strategy.
3. Table 4 focuses on 7B/8B models. Experiments on other models with different size (for example, similar in size to phi-2) are needed to further establish generalizability.
4. The Win-Tie evaluation's 200-sample size, while understandable due to cost, might compromise the solidity and statistical significance of the results.

**Strengths Contributions:**

1. PrefCleanBench is the first comprehensive benchmark for evaluating preference data cleaning methods specifically for LLM alignment.
2. The benchmark covers 13 diverse cleaning methods across three paradigms. The selection of multiple standard datasets, various LLM backbones, and several alignment algorithms, ensure robust and generalizable findings.

---

> ### Author Rebuttal · Authors · 2025-07-28
>
> We sincerely appreciate your feedback and insightful comments! We address the questions and comments below in detail.
>
> > Computational resources and trade-off between effectiveness and cost
>
> Thank you for raising this excellent question. Below we summarize all computational resources/API costs for each data cleaning approach, using the Anthropic-HH dataset (N=160k) as reference.
>
> - LLM-Judge-R/LLM-Judge-R: Given the GPT-4o API pricing ($2/1M input tokens and $8/1M output tokens), the total API cost on Anthropic-HH is approximately 350USD (<1000 input tokens and <20 output tokens for each data point).
> - RwGap-R/RwGap-F: Training 8 DPO models takes under 12 hours on 4xH200 GPUs. Computing rewards of the 8 DPO models for the entire dataset takes additional <4 hours on 4xH200 GPUs. In total, it takes less than **16 hours** on 4xH200 GPUs to clean the dataset.
> - VoteAll-R/VoteAll-F/VoteMaj-R/VoteMaj-F: Each reward model takes <1 hour on 4xH200 GPUs to compute reward for the entire dataset. In total, it takes less than **6 hours** on 4xH200 GPUs to clean the dataset.
> - Tag-Cmp/Tag-Div: Generate tags and clean the full dataset takes **>24 hours** using HuggingFace's `AutoModelForCausalLM`. The process could be significantly faster with optimized backends like vLLM.
> - IFD-R/IFD-Gap-R/IFD-Gap-F: It takes less than **6 hours** to compute IFD score with Llama3-8B on 4 H200 GPUs for the entire dataset.
>
> Overall, we find that VoteMaj-R and IFD-Gap-R offer strong trade-offs between cleaning effectiveness and computational efficiency. We will include this cost-performance breakdown in the revised paper to guide practitioners in choosing between methods based on available resources.
>
> Thanks again for encouraging us to clarify this important aspect.
>
> > Human evaluation
>
> Thank you for raising this important point. We agree that relying on a single model like GPT-4o may introduce bias. To address this, we have taken two steps:
>
> 1. **We conducted human evaluation, as suggested**. Specifically, we randomly sample a subset of data points from the Anthropic-HH dataset, and compare the responses generated by Llama3-8B trained with DPO on the original dataset and on the dataset cleaned by VoteMaj-R. We conduct both human annotation and LLM-judge with GPT-4o, and compute the Cohen's kappa inter-annotator agreement score. The result shows a significant Cohen's kappa value, supporting the reliability of GPT-4o as a practical proxy.
> 2. In our manuscript, we also complemented the GPT-4o-based WinTie rate with gold reward models for evaluation (L200-L202). To ensure robustness, we repeated the evaluation protocol using **multiple alternative gold reward models** (see **Appendix F** for details), which helps mitigate potential bias in any single model.
>
> We will update the paper to reflect these efforts and clarify our evaluation strategy accordingly.
>
>
> > Experiments on other models with different size
>
> We conduct experiments on Llama3.2-1B. The results in the tables below show a similar trend to the result of Phi-2, suggesting the generalizability across models.
>
> #### WinTie
> |Model|Approach|Anthropic-HH|UltraFeedback|SafeRLHF|HelpSteer2|
> |---|---|---|---|---|---|
> |Phi-2|VoteMaj-R|0.590|0.650|0.715|0.585|
> |Phi-2|Tag-Cmp|0.395|0.605|0.780|0.645|
> |Llama3.2-1B|VoteMaj-R|0.655|0.625|0.735|0.590|
> |Llama3.2-1B|Tag-Cmp|0.580|0.515|0.665|0.600|
>
> #### Avg. Rwd
> |Model|Approach|Anthropic-HH|UltraFeedback|SafeRLHF|HelpSteer2|
> |---|---|---|---|---|---|
> |Phi-2|Vanilla|5.626|2.712|7.570|4.492|
> |Phi-2|VoteMaj-R|6.287|2.644|9.204|4.187|
> |Phi-2|Tag-Cmp|4.382|2.767|5.511|4.338|
> |Llama3.2-1B|Vanilla|4.441|3.031|4.720|4.012|
> |Llama3.2-1B|VoteMaj-R|5.857|3.081|7.431|3.891|
> |Llama3.2-1B|Tag-Cmp|4.485|2.569|6.043|3.894|
>
> > Sample size in evaluation
>
> Thank you for pointing this out. We agree that larger sample sizes are always preferred for statistical robustness. However, we note that our 200-prompt Win-Tie evaluation follows the same scale commonly used in recent alignment literature, including prior works such as[1, 2], which only evaluated on 100 samples.
>
> To further bolster the robustness of our evaluation, we also report **average reward scores on the full test set**, which complement Win-Tie results and provide an additional, statistically stable signal.
>
> Moreover, to directly address this concern, we conducted an additional large-scale Win-Tie evaluation on 1,000 samples, comparing DPO models trained on Llama3-8B using the original data versus data cleaned by VoteMaj-R. As shown below, the trends remain similar in most cases, reinforcing the statistical significance of our findings:
>
> |Sample Size|Approach|Anthropic-HH|UltraFeedback|SafeRLHF|
> |---|---|---|---|---|
> |200|VoteMaj-R|0.705|0.650|0.770|
> |1000|VoteMaj-R|0.679|0.632|0.770|
>
> ---
> [1]: Li, Ming et al., From quantity to quality: Boosting LLM performance with self-guided data selection for instruction tuning. 2024.
>
> [2]: Wang, Binghai et al., Secrets of rlhf in large language models part ii: Reward modeling. 2024.

---

> > ### Comment · Reviewer_1XUo · 2025-08-04
> >
> > Thanks for your hard work. The detailed experiments solve all my concerns. I will raise my score to 4. Good luch for the rest process!

---

> > > ### Author Response · Authors · 2025-08-04
> > >
> > > We are glad the response addressed your concerns, and once again, thank you for your support and insightful comments!

---

### Official Review · Reviewer_Y63P · 2025-06-15

**Rating:** 5
**Confidence:** 2

**Summary:**

The paper presents a comprehensive benchmark for systematically evaluating 13 methods aimed at cleaning noisy human preference data—a critical step in aligning large language models (LLMs) with human values. The experiments yield several key findings: removing noisy data is more effective than flipping labels, multi-judge strategies outperform single-judge approaches, and performance generalizes well across models and training setups. The well-structured benchmark and its insights offer valuable contributions to the field of preference data cleaning.

**Additional Feedback:**

As I am not deeply familiar with evaluating papers in the benchmark track, I provide this review with low confidence. That said, I believe the paper addresses an important problem—preference data noise—which is a major bottleneck in aligning large language models (LLMs), and it offers several interesting and valuable findings. While I feel the task lacks a concrete definition of what constitutes an inconsistent preference, and that many benchmarks in this area tend to be heuristic in nature, this may be outside the main focus of the paper. Nonetheless, I would encourage the authors to clarify this aspect to strengthen the overall framing.

**Dataset Code Accessibility:**

Yes

**Dataset Code Comments:**

Code and pipeline are open-sourced. The protocol is modular and designed for community adoption.

**Ethical Considerations:**

No, there are no or only very minor ethics concerns

**Final Justification:**

As the authors have detailed explained the mathmatical formulation of the preference data cleaning, I have a slightly stronger confidence to support my score of acceptance.

**Limitations Weaknesses:**

The paper lacks human evaluation to establish ground truth for data cleaning, which could significantly strengthen the analysis.

The results would benefit from greater interpretability—for example, explaining why certain LLM-as-a-Judge methods underperform in specific settings.

A more detailed introduction to the task of preference data cleaning is needed, particularly a mathematically grounded definition of what constitutes preference noise.

**Strengths Contributions:**

This paper tackles the challenge of noisy preference data—a major bottleneck in aligning large language models (LLMs)—and offers modular open-source code along with coverage of widely-used datasets and models. It demonstrates strong experimental depth, including multi-dimensional generalization, ablations on filtering thresholds, and comparisons across multiple reward models. The key findings are both interpretable and practically valuable, offering new insights into effective strategies for preference data cleaning.

---

> ### Author Rebuttal · Authors · 2025-07-28
>
> We sincerely appreciate your feedback and insightful comments! We address the questions and comments below in detail.
>
> > Human evaluation
>
> Thank you for raising this important point. We fully agree that incorporating human evaluation can significantly strengthen the analysis.
>
> In response, we have conducted a human evaluation study as suggested. Specifically, we sample 50 data points from the Anthropic-HH dataset, and compare the responses generated by Llama3-8B trained with DPO on the original dataset and on the dataset cleaned by VoteMaj-R. We conduct both human annotation and LLM-judge with GPT-4o, and compute the Cohen's kappa inter-annotator agreement score. The result shows a high Cohen's kappa value, suggesting a significant agreement between human judgments and GPT-4o assessments.
>
> To further address potential bias from relying on only human or a single model, we also complemented our evaluation with a suite of alternative reward models (see Appendix F for details), demonstrating consistent trends across evaluation choices.
>
> We will update the paper to reflect these efforts and clarify our evaluation strategy accordingly.
>
> > Why certain LLM-as-a-Judge methods underperform
>
> This is another insightful question. One reason is that the LLM-as-a-Judge methods (LLM-Judge-R/LLM-Judge-F) rely only on a single model to identify unreliable. As we've explained in Section 5.1, such a setting may lead to biasing LLMs toward certain/unexpected behaviors. To better illustrate this idea, we analyze 50 data points sampled from the Anthropic-HH dataset that are marked as unreliable by LLM-Judge but reliable by VoteMaj, and another 50 data points that are marked as unreliable by VoteMaj but reliable by LLM-Judge. We found that the discrepancy between LLM-Judge and VoteMaj usually happens when the two response candidates have a similar quality. Specifically, when both responses were suggesting harmful behaviors, since LLM-Judge is forced to decide which response is better, it has around a 1/2 probability of choosing the "chosen one" and keeping the data point in the dataset. In contrast, the decision of VoteMaj is made by multiple models, so these data tend to get mixed votes and are more likely to be removed. Since these low-quality data are harmful for aligning LLMs, training on them will degrade the performance.
>
> > Mathematical definition of preference data cleaning
>
> Denote $\mathbb{P}^+$ be a distribution of high-quality preference data, in which each data point $d:=(x,y_1,y_2,l)$ consists of a high-quality prompt $x$, two response candidates $y_1, y_2$, and a reliable label $l\in\{\succ, \prec\}$, where $\succ$ means $y_1$ is better than $y_2$ and $\prec$ means $y_2$ is better than $y_1$. Also denote $\mathbb{P}^-$ be a distribution of low-quality preference data, in which each data point $d$ has the same structure $(x, y_1, y_2, l)$, while the prompt $x$ (and/or both response candidates $y_1,y_2$) are low quality and/or the label $l$ is unreliable (e.g., $l:=\succ$ when $y_2$ is better than $y_1$). We assume a noised preference dataset $\mathcal{D}$ consists of data sampled from a mixture distribution $\mathbb{P}=(1-\alpha)\mathbb{P}^++\alpha\mathbb{P}^-$. The task of preference data cleaning is to remove or correct data points in $\mathcal{D}$ that are sampled from $\mathbb{P}^-$ such that the cleaned dataset $\mathcal{D}'$ contains data purely sampled from $\mathbb{P}^+$. We will revise our manuscript with a more precise definition—thanks again for your valuable suggestion!

---

> > ### Comment · Reviewer_Y63P · 2025-08-04
> >
> > Thanks for the rebuttal. My concerns has been addressed and I slight raise my confidence and keep my score.

---

> > > ### Author Response · Authors · 2025-08-04
> > >
> > > We are glad the response addressed your concerns, and once again, thank you for your support and insightful comments!

---

### Official Review · Reviewer_rnUU · 2025-06-26

**Rating:** 4
**Confidence:** 2

**Summary:**

This paper presents a comprehensive benchmark for evaluating 13 existing preference data cleaning methods in the context of Large Language Model (LLM) alignment. The authors conduct a systematic evaluation across four widely-used preference datasets, multiple base LLMs, and various preference optimization algorithms. The primary claimed contributions are:

1. proposing a standardized protocol for comparing different data cleaning strategies;

2. demonstrating through extensive experiments that both the identification and treatment of unreliable data are critical, with a key finding that using multiple judges to identify and subsequently remove unreliable data generally outperforms using a single judge or flipping labels;

3. highlighting that data quality should be prioritized for effective alignment;

4. providing an open-source, modular implementation of the evaluated methods.

**Additional Feedback:**

Overall, I think this is an interesting paper. If the authors could address my concerns during the rebuttal stage, I would be happy to raise my score.

**Dataset Code Accessibility:**

Yes

**Dataset Code Comments:**

The authors provide an anonymized link to a code repository in the abstract, promising to release modular implementations of all evaluated methods. The datasets used are publicly available and properly cited (Sec 4.1). This provides a solid foundation for reproducibility and follow-up research.

**Ethical Considerations:**

No, there are no or only very minor ethics concerns

**Final Justification:**

I am convinced of the paper's contribution, but uncertain about how the public will perceive and interpret the contributions of this paper if it is accepted.

I am torn between 3 and 4, and so opted for 4 while lowering confidence.

**Limitations Weaknesses:**

1. Severe Lack of Novelty in Conclusions and Insights: The paper's primary weakness is that its core conclusions are substantially anticipated and discussed within its own cited literature, critically diminishing the work's originality and impact. For example, the main finding—that using multiple judges to identify and remove unreliable data is superior—is heavily foreshadowed by the reference [81], which already advocated for this strategy and cited prior work as evidence of its effectiveness.

2. Insufficient Novelty as a "Benchmark" Contribution: For a submission to the "Datasets and Benchmarks" track, the work lacks the expected level of novelty in its methodological contribution. It relies entirely on existing public datasets (Sec. 4.1) and standard evaluation metrics (Sec. 4.3) without proposing new, challenging data or innovative metrics tailored to the nuances of data cleaning. Consequently, the work functions more as a large-scale comparative study than a novel benchmark that establishes new evaluation paradigms for the community.

3. Insufficient Depth of Analysis: Even as a large-scale empirical study, the paper's analysis remains somewhat superficial. It often stops at reporting performance differences without delving deeply into the underlying reasons. For example, beyond referencing the "six sources of unreliability" from [81] (lines 232-237), the paper misses the opportunity to leverage its own extensive data to provide new, fine-grained mechanistic explanations for why certain cleaning strategies succeed or fail under different conditions.

**Strengths Contributions:**

The paper focus on a timely and critical problem in LLM alignment concerning the impact of preference data quality. Its primary contribution is a large-scale empirical study that systematically evaluates 13 existing cleaning methods across a wide range of datasets, models, and optimization algorithms (Sec. 5). This extensive evaluation, grounded in a clear protocol (Figure 1) and complemented by the release of open-source modular code (Abstract), provides a potentially useful and reproducible resource for the research community. The paper is well-written and clearly organized, making its methodology and results easy to follow.

---

> ### Author Rebuttal · Authors · 2025-07-28
>
> We sincerely thank the reviewer for the thorough and constructive feedback. Below, we address the concerns in detail.
>
>
> ---
> _A1. Clarification on the novelty of conclusion and insights_
>
> We appreciate the reviewer’s concern and would like to offer important clarifications. First, reference [81] is a position paper, which **does not provide any comparative analysis across cleaning strategies**, and thus makes no claim that "using multiple judges to identify and remove unreliable data _is superior_”. In contrast, our submission is **the first to systematically operationalize, benchmark, and quantify across 13 cleaning strategies**, 4 diverse preference datasets, and 8 preference optimization algorithms—all within a unified, reproducible evaluation framework. Our work constitutes a substantial leap from this position paper, and is the first to draw conclusion that using multiple judges to identify and remove unreliable data is superior. In other words, **this conclusion is indeed new to our work**, not [81].
>
> Additionally, our benchmark reveals other novel findings that significantly go beyond prior work [81], as endorsed by other reviewers
>
> > "_The key findings are both interpretable and practically valuable, offering new insights into effective strategies for preference data cleaning_." (Reviewer Y63P)
>
> - (1) We show that removal consistently outperforms label flipping (Table 1), which had not been systematically tested before;
> - (2) Our benchmark uncovers a non-trivial interaction between data cleaning and preference optimization algorithms (Section 5.2). This has not been studied or discovered before.
> - (3) Our benchmark also demonstrates that heuristic methods like Tag-Cmp, despite lacking judge supervision, can rival judge-based methods in certain regimes.
>
> These insights, enabled by the **breadth and rigor of our evaluation**, provide novel guidance to practitioners and researchers building data-centric alignment pipelines—**moving the field beyond isolated anecdotal evidence toward systematic and reproducible understanding, making the future development of data cleaning approaches better grounded in evaluation**.
>
> ---
> _A2. Benchmark contribution_
>
> We appreciate the opportunity to clarify the novelty and significance of our benchmark. While new datasets/metrics are certainly valuable, they are not the sole or primary requirement for evaluating papers in the NeurIPS Datasets and Benchmarks track:
>
> > _Novelty may come from the data, the task, the protocol, or the evaluation; not all are required. Papers that introduce systematic evaluation of existing methods, even on existing datasets, are considered in scope if they provide new insight or reusable tools._
>
>
> As per official NeurIPS D&B guidelines, submissions are encouraged with topics on
>
> - Data-centric AI methods and tools, e.g., to measure and improve data quality or utility, or studies in data-centric AI that bring important new insight.
> - Frameworks for responsible dataset development, audits of existing datasets, identifying significant problems with existing datasets and their use
>
>
> Our submission fits this scope precisely: it addresses a critical and previously underexplored gap in the alignment field by designing and implementing the first benchmark for preference data cleaning. This is a highly impactful area of alignment research, and **no prior work has established a standardized protocol that enables systematic, apples-to-apples comparison across cleaning strategies, datasets, models, and optimization algorithms.**
>
> Notably, the significance of our benchmark was recognized by multiple reviewers:
>
> > “_A key contribution is the development of a standardized evaluation protocol for assessing preference data cleaning methods. This is highly valuable for practitioners seeking to understand which approaches most effectively enhance LLM alignment and the generalizability of these methods._”
> > — *Reviewer t6r1*
>
> > “_The paper presents a comprehensive benchmark for systematically evaluating 13 methods aimed at cleaning noisy human preference data...The well-structured benchmark and its insights offer valuable contributions to the field of preference data cleaning_.”
> > — *Reviewer Y63P*
>
> We also wish to emphasize that our work goes beyond a simple comparative study in several important ways:
>
> 1. **Our modular integration of 13 diverse cleaning approaches is both new and non-trivial effort**. We standardize and implement a wide range of approaches—that were previously fragmented or undocumented—under a consistent pipeline that supports plug-and-play evaluation across datasets, alignment objectives, and model backbones. This effort required substantial engineering and design to ensure comparability, making it a valuable resource for the community.
>
> 2. Our protocol unifies multiple axes of evaluation—alignment effectiveness, generalizability across optimization algorithms and LLM backbones, and ablation over filtering thresholds—thereby **laying the groundwork for future studies to evaluate data-centric techniques in a reproducible and extensible way**.
> 3. Our benchmark surfaces previously unknown interactions between cleaning methods and alignment algorithms (Sec. 5.2), and identifies concrete best practices that had not been empirically validated at this scale.
>
>
> We hope this clarifies that our submission is well within the intended scope of the NeurIPS D&B track and makes a novel and timely contribution to the study and practice of alignment.
>
> ---
> _A3. Depth of analysis_
>
> We appreciate the reviewer’s suggestion to provide more fine-grained mechanistic explanations. Our primary goal was to establish a comprehensive, standardized benchmark for evaluating cleaning strategies, which required evaluating a broad set of 13 approaches across diverse settings. That said, we absolutely agree that a deeper analysis of why some methods succeed under specific conditions is valuable.
>
> To that end, **our paper already includes several such analyses**. In **Section 5.2**, we examined interactions between cleaning methods and optimization algorithms. For instance, we attributed VoteMaj-R's success with AOT/ORPO to their distribution-aware nature, and Tag-Cmp’s effectiveness with KTO/rDPO to their robustness to noise (**L266-280**). These observations go beyond performance reporting and uncover algorithm-specific insights, which are also recognized by Reviewer t6r1
>
> > "_The authors conduct comprehensive experiments under varied settings and provide an **in-depth analysis** of each method's performance._"
>
> Nonetheless, we agree that this area deserves further exploration. **In response to your suggestion, we conducted an additional qualitative study** comparing LLM-Judge and VoteMaj. Specifically, we examined 50 data points marked as unreliable by LLM-Judge but retained by VoteMaj, and another 50 marked unreliable by VoteMaj but kept by LLM-Judge. We found that discrepancies often occur when both candidate responses are similarly low in quality—e.g., both suggest harmful behavior. In these borderline cases, LLM-Judge is forced to make a binary decision and may randomly retain such data, whereas VoteMaj, aggregating over multiple model opinions, is more likely to flag the disagreement and remove the example. Since training on such harmful examples degrades alignment, this helps explain why VoteMaj tends to outperform LLM-Judge empirically.
>
> Additionally, we analyzed why VoteMaj-R and VoteMaj-F exhibit a significant performance gap on HelpSteer2. To do so, we examined 50 data points marked as unreliable by VoteMaj, as well as another 50 data points that were retained. We observed a significant gap in the quality of the input prompt between the unreliable and retrained data. Specifically, the retrained data tends to have a prompt with a clear instruction or a specific question, leading to high-quality response candidates and reliable preference annotations. In contrast, a large amount of unreliable data marked by VoteMaj has low-quality prompts. For example, simply greeting LLMs, posting a vague question, or asking LLMs to generate a list of product descriptions without providing any data. LLMs prompted on them usually generate responses that are generic or hallucinated. In this case, VoteMaj-F, i.e., flipping the labels of unreliable data, can not mitigate the unreliability because it is due to the prompt. In fact, flipping the labels even degrades the performance because some marked data have a correct label. On the other hand, VoteMaj-R removes all the unreliable data, cleaning up data with a low-quality prompt and preventing the risk of wrongly correcting labels. This observation helps explain why removing is better than flipping when using VoteMaj.
>
> We will incorporate these additional analyses in our revised manuscript. Thanks again for your thoughtful comment!

---

### Official Review · Reviewer_t6r1 · 2025-06-27

**Rating:** 5
**Confidence:** 4

**Summary:**

Human feedback for aligning large language models (LLMs) with human preferences is often noisy or inconsistent. These noisy feedbacks can compromise the effectiveness of alignment. While various methods have been proposed to mitigate the impact of such noise, there remains a lack of systematic evaluation. The authors propose the first comprehensive benchmark to assess 13 preference data cleaning methods in the context of LLM alignment. They also present a standardized evaluation protocol for consistent and rigorous comparison across methods.

**Additional Feedback:**

Some definitions are missing or insufficiently explained in the paper. For example, in Lines 134–135, the authors state that “*p*% of the data with the smallest reward gap are considered to have incorrect labels,” but the term reward gap is not clearly defined in the main text. The term *reward gap* appears to admit both positive and negative values, as shown in Figure 2. While this is reasonable, the paper should explicitly define how the reward gap is computed (i.e., the order of subtraction).

**Dataset Code Accessibility:**

Yes

**Dataset Code Comments:**

The code is available at the provided URL, and detailed instructions are included for the reproduction of the benchmark.

**Ethical Considerations:**

No, there are no or only very minor ethics concerns

**Final Justification:**

The authors addressed my concerns. Thus, I maintain my positive score.

**Limitations Weaknesses:**

1. The paper does not evaluate how data cleaning methods behave when the feedback is already clean (i.e., an ideal or low-noise setting). It remains unclear whether these cleaning methods could degrade alignment performance in such cases. This paper does not provide a controlled experiment with clean feedback or has a discussion on it.
2. As shown in Table 2, the proportion of data removed significantly affects alignment performance. However, the paper does not provide practical guidance on how to select this proportion in real-world scenarios where ground truth is unavailable.

**Strengths Contributions:**

1. The paper is well-written, clearly structured, and easy to follow. The authors effectively communicate the motivation, methodology, and findings.
2. A key contribution is the development of a standardized evaluation protocol for assessing preference data cleaning methods. This is highly valuable for practitioners seeking to understand which approaches most effectively enhance LLM alignment and the generalizability of these methods.
3. The benchmark includes 13 representative data cleaning methods, including three major categories: (1) LLM-as-a-judge methods, (2) reward model-based methods, and (3) heuristic-driven methods.
4. The authors conduct comprehensive experiments under varied settings and provide an in-depth analysis of each method's performance. This includes evaluations across multiple datasets and preference optimization algorithms, offering valuable insights into the strengths and limitations of existing methods.

---

> ### Author Rebuttal · Authors · 2025-07-28
>
> We sincerely appreciate your feedback and insightful comments! We address the questions and comments below in detail.
>
> > Data cleaning on cleaned dataset
>
> Thank you for raising this important point. While our primary focus was on real-world preference datasets where noise is pervasive, we fully agree on the need for controlled evaluation under clean conditions.
>
> As suggested, we have conducted additional experiments to understand this aspect. In practice, it is extremely challenging to identify or verify a perfectly clean dataset. We simulate a proxy "clean" setting by filtering the dataset using VoteMaj-R, one of the most reliable judge-based strategies in our benchmark. We then apply Tag-Cmp, a representative method, on top of this proxy "clean" dataset. The results (included below) show that Tag-Cmp preserves the performance in most cases, suggesting that additional cleaning does not significantly harm alignment in cleaner regimes.
>
> #### WinTie
> |Approach|Anthropic-HH|UltraFeedback|SafeRLHF|HelpSteer2|
> |---|---|---|---|---|
> | "clean"|0.705|0.650|0.770|0.750|
> | "clean"+Tag-Cmp|0.695|0.620|0.800|0.740|
>
> #### Avg. Rwd
> |Approach|Anthropic-HH|UltraFeedback|SafeRLHF|HelpSteer2|
> |---|---|---|---|---|
> |"clean"|7.287|4.253|8.478|6.834|
> |"clean"+Tag-Cmp|7.444|3.613|9.364|6.801|
>
>
> > How to select the removal proportion
>
> We agree that selecting the right proportion of removal data is essential and challenging, which we believe is a limitation of score-based cleaning methods. Unlike voting, they don't have a clear-cut way to remove low-quality data. Existing works [1, 2] didn't explain how to find the optimal ratio, motivating us to explore the impact of different removal proportions (10%-40%). As a benchmark paper, our goal is not to propose a way to find an optimal removal ratio but to compare the effectiveness of different approaches. While we believe that selecting the right proportion of removal data is a valuable future work, and a plausible direction may be examining the distribution of scores (e.g., reward gap) to identify the cutting point.
>
> > Definition of reward gap
>
> Given a pairwise preference data $d=(x, y_c, y_r)$, where $x$ is the prompt, $y_c$ is the chosen response, and $y_r$ is the rejected response, the reward gap w.r.t. a reward model $r$ is defined as
> $$\mathrm{RwGap}_r(d):=r(x,y_c)-r(x,y_r).$$
> We will add this definition to the revised manuscript. Thanks again for your careful read！
>
> ---
> [1]: Li, Ming et al., From quantity to quality: Boosting LLM performance with self-guided data selection for instruction tuning. 2024.
>
> [2]: Wang, Binghai et al., Secrets of rlhf in large language models part ii: Reward modeling. 2024.

---

> > ### Comment · Reviewer_t6r1 · 2025-08-04
> >
> > Thanks for the rebuttal. My concerns are addressed, so I maintain my score. The experiments on the proxy "clean" dataset are recommended to be included in the final paper.

---

> > > ### Author Response · Authors · 2025-08-04
> > >
> > > We are glad the response addressed your concerns, and once again, thank you for your support and insightful comments!

---

### Comment · Area_Chair_UFFa · 2025-08-06
**Please engage in the reviewer-author discussion.**

Dear Reviewers,

Thank you for providing the initial reviews. Please respond to the authors' rebuttal and engage in the reviewer-author discussion if you haven't done so.

Best,

AC

---

### Decision · Program_Chairs · 2025-09-18

**Decision:**

Accept (poster)

**Comment:**

This paper addresses a timely and important problem in LLM alignment by focusing on the often-overlooked issue of preference data quality. It makes a good contribution by introducing a benchmark that systematic analyzes 13 data cleaning methods with a unified evaluation protocol. The empirical study is thorough, spanning diverse datasets, models, and alignment algorithms, and is accompanied by modular open-source code to support reproducibility. The paper is clear, rigorous, and impactful, with findings that will be valuable both for researchers and practitioners in LLM alignment. Most of reviewers' concerns have been addressed and all reviewers gave this paper positive ratings. Overall, this is a good and timely contribution.